# Cattle Body Size Measurement Based on DUOS–PointNet++

**DOI:** 10.3390/ani14172553

**Published:** 2024-09-02

**Authors:** Zhi Weng, Wenzhi Lin, Zhiqiang Zheng

**Affiliations:** 1State Key Laboratory of Reproductive Regulation and Breeding of Grassland Livestock, Inner Mongolia University, Hohhot 010021, China; wzhi@imu.edu.cn (Z.W.); 32256136@mail.imu.edu.cn (W.L.); 2College of Electronic Information Engineering, Inner Mongolia University, Hohhot 010021, China; 3Research Base for Dairy Farming Engineering and Full Mechanization of Equipment, Ministry of Agriculture and Rural Affairs, Inner Mongolia Agricultural University, Hohhot 010018, China

**Keywords:** body size measurement, point clouds, PointNet++

## Abstract

**Simple Summary:**

In intelligent agriculture, non-contact, automatic body size measurement is widely used for livestock. However, the common measurement methods based on the whole point cloud are complex and prone to errors because of the positions of the cattle. To measure the body sizes of livestock more accurately, a cattle body measuring system is proposed. The system includes a new algorithm called dynamic unbalanced octree grouping (DUOS), based on PointNet++, and an efficient method of body size measurement based on segmentation results. The network divides the cow into the following seven parts: body length, withers height, hip height, thoracic circumference, abdominal circumference, and cannon circumference. Compared with some of the other models, the system has higher segmentation accuracy and lower measurement errors. The system can be widely applied in the non-contact body measurement of livestock; in addition, it can increase efficiency and reduce costs. It also has broad prospects in the intelligent livestock industry.

**Abstract:**

The common non-contact, automatic body size measurement methods based on the whole livestock point cloud are complex and prone to errors. Therefore, a cattle body measuring system is proposed. The system includes a new algorithm called dynamic unbalanced octree grouping (DUOS), based on PointNet++, and an efficient method of body size measurement based on segmentation results. This system is suitable for livestock body feature sampling. The network divides the cow into seven parts, including the body and legs. Moreover, the key points of body size are located in the different parts. It combines density measurement, point cloud slicing, contour extraction, point cloud repair, etc. A total of 137 items of cattle data are collected. Compared with some of the other models, the DUOS algorithm improves the accuracy of the segmentation task and mean intersection by 0.53% and 1.21%, respectively. Moreover, compared with the manual measurement results, the relative errors of the experimental measurement results are as follows: withers height, 1.18%; hip height, 1.34%; body length, 2.52%; thoracic circumference, 2.12%; abdominal circumference, 2.26%; and cannon circumference, 2.78%. In summary, the model is proven to have a good segmentation effect on cattle bodies and is suitable for cattle body size measurement.

## 1. Introduction

In evaluating cattle indicators, body size parameters are essential [1]. In the past, livestock data were measured manually [2,3], resulting in detrimental effects related to animal welfare protection [4]. Non-contact measurement is the key aspect of the integration of intelligence into animal husbandry, and previous studies have reported many successful cases [5].

A two-dimensional (2D) image acquisition measurement system was established to calculate the size data of cows [6]. A foreground extraction algorithm based on SLI and FCM and a point extraction measurement algorithm were proposed to extract and calculate the body size of sheep [7]. 

However, the two-dimensional image processing method requires strict calibration and only measures partial body sizes accurately, while lacking precision for three-dimensional (3D) spatial data. To solve these problems, several researchers have used the Kinect series of depth cameras for depth data collection [8,9,10]. In [11,12], multiple depth cameras were used to obtain the point cloud and perform 3D reconstruction, which is a common method. Subsequently, the Poisson surface reconstruction algorithm was used to complete the measurements [13]. The symmetry properties of unilateral or bilateral point clouds have been applied [14,15]. With registration using a cuboid and the polar coordinate transformation algorithm, it was possible to calculate the parameters more accurately [16]. Kinect v4 and YOLOv5 were used to remove cattle from the environment to measure body size [17]. Subsequently, fusion processing methods based on 2D and 3D data were proposed [18]. Depth-based camera features were used to classify and transform a pig’s posture based on its skeleton [19]. An acquisition method with a single depth camera placed on a rod was proposed [20]. In addition, an improved segmentation model to measure pig body size using part of the body was proposed [21]. Machine learning algorithms were used to measure the body size and weight of cows [22,23]. Depth images were used for cattle body condition scoring [24].

At present, most of cattle body size measurements based on a point cloud use the complete cattle point cloud after registration. The relationship between body size parameters and genome-wide association studies was investigated [25]. A measurement method involving the extraction of key points from the point cloud of the entire body of the cow was used to measure body size [26]. 

The advantage of this method is that it reduces the number of data processing steps and ensures the continuity of the data. However, due to the large number of points in the collected and processed cow’s complete point cloud, along with the varying positions and occlusions of different body parts, considerable time is consumed in model calculation. This also results in interference from the point clouds originating from other parts of the body, resulting in the positioning error of key points. Moreover, for nonrigid objects such as cattle, the traditional point cloud segmentation network model causes inaccurate segmentation, low evaluation parameters, poor robustness, and other problems.

To solve these problems, an improved deep learning model based on the improved octree algorithm and the segmentation model of PointNet++ is proposed. The proposed method can segment each part of the cattle point cloud, then locate the key points of cattle body measurement according to the geometric features, effectively improving the accuracy of the subsequent body measurement of cattle. Simultaneously, a measurement method is proposed to measure cattle body size based on the segmented body of the cow. The proposed method makes it easier to extract the key points of measurement, achieving higher accuracy.

## 2. Materials and Methods

In this study, no animals were used or cared for. Moreover, no animal care or ethics committee approval was required because the data used in the study were obtained from routine cattle herd management practices.

### 2.1. Data Collection

The cattle data collection was completed in 2023 in the pasture of Shengquan Animal Husbandry Co., Ltd., located in Wengniute Banner, Chifeng City, Inner Mongolia Autonomous Region, China. Kinect V2 was chosen because it is one of the most widely used cameras. It is able to cope with the body size measurement requirements in most cases. Three KinectV2 cameras were installed directly above and on the left and right sides, 1.2 m from the channel. When a target cow entered, each camera detected and collected the data synchronously, and the data were then saved to the computer. A field image is shown in Figure 1.

### 2.2. Point Cloud Preprocessing

Following the storage of the 3D cattle body point cloud data, RANSAC [27] and ICP [28] were used to register the data as a complete point cloud. The algorithms are simple, commonly used, and relatively effective registration algorithms. Filtering is used to remove the background point clouds, and part of the background point cloud noise is removed using CloudCompare software 2. 13. alpha. Outlier points are removed through radius filtering. The point cloud preprocessing is shown in Figure 2.

The point cloud data are normalized in the model before sampling, and the specific steps are as follows:Take the average of each row of data in the point cloud to obtain the centroids;Translate the point cloud so that the centroid is located at the origin;Calculate the maximum norm of the point cloud data and normalize the data according to the maximum norm, so that the range of the point cloud data is between [−1, 1].

### 2.3. Data Marking

According to the real requirements of cattle body size measurement, the whole point cloud of the cow’s body is segmented to facilitate the follow-up body size measurement. Given that the head and tail of the cow play a minimal role in body size measurement in this study, they are excluded from the point cloud. The multiple complete point clouds of the cow are divided into several parts using CloudCompare software. Specifically, the legs and body of the cow are separated into seven distinct parts using CloudCompare, with each part being labeled accordingly. In addition, the cow’s legs are subdivided into four parts labeled 0–3, while the body of the cow is subdivided into three parts labeled 4–6, representing the front, middle, and back.

### 2.4. Improved Model

#### 2.4.1. Overall Framework

Previous studies have proposed using the PointNet [29] and PointNet++ [30] methods to process point clouds and make improvements. The PointNet++ model uses the farthest point sampling (FPS) algorithm, which captures rigid deformation well. However, it does not perform well for nonrigid objects such as cattle. Therefore, in this study, an improved dynamic unbalanced octree sampling method is proposed to replace FPS in order to improve cattle segmentation using PointNet++. The specific architecture of the improved model is shown in Figure 3. 

The processed cattle point cloud data, including the total number of data, the number of single data, and the data dimensions, are input;The cow’s body point cloud is sampled and grouped based on the dynamic unbalanced octree grouping (DUOS) algorithm, and the maximum point that the number of leaf nodes can accommodate is *n*, where $n_{1}$ is defined as the number of sampling centers;After sampling and grouping, the local point cloud is processed using PointNet;Steps 2 and 3 are repeated;The feature vectors are processed using PointNet.

The following inverse distance-weighted interpolation (Formula (1)) is used to calculate the weight according to the reciprocal of the distance; the attribute value of the unknown position is estimated, and the data are obtained. Hence, the characteristics of the point cloud ignored in the downsampling process and the labels of each point in the complete point cloud are obtained.
(1)$$f^{\left(j\right)}\left(x\right)=\left\{\begin{array}{c} \frac{\sum_{i=1}^{k}\omega_{i}\left(x\right)f_{i}^{\left(j\right)}}{\sum_{i=1}^{k}\omega_{i}\left(x\right)},\;if\;d{\left(x,x_{i}\right)}^{p}\neq 0\;for\;all\;i\\ f_{i}^{\left(j\right)}\;\;\;\;,\;if\;d{\left(x,x_{i}\right)}^{p}\neq 0\;for\;all\;i \end{array}\right.\;\mathrm{where}\;\omega_{i}\left(x\right)=\frac{1}{d{\left(x,x_{i}\right)}^{p}},j=1,\cdots ,C$$

6.Seven classification values are obtained after processing. The maximum value is taken as the output result to complete the target cow’s point cloud segmentation. The cross-entropy loss function (Formula (2)) is used as the loss function for the network.
(2)$$Loss=-\frac{1}{N}\sum_{i=0}^{N-1}\sum_{k=0}^{N-1}y_{i,k}\ln p_{i,k}$$
where $y_{i,k}$ means the true label, K is the number of label values, and $p_{i,k}$ is the probability of $y_{i,k}$. The algorithm flow chart is shown in Figure 4.

#### 2.4.2. Dynamic Unbalanced Octree Grouping

The proposed algorithm (1) constrains the number of points allowed in a leaf node instead of setting layers and (2) adds density determination to the sampling point selection in each leaf node. 

Traditional octree segmentation is based on the number of layers, but cattle point clouds are characteristically dense in the body and sparse in the limbs. If the number of layers is fixed, the point cloud of the legs cannot properly fill the leaf node, which affects the feature weights in subsequent sampling. The DUOS algorithm determines whether to continue layering by controlling the amount of data in each leaf node and then constructs a non-balanced octree that better deals with the density of the point cloud in different regions.

To avoid noise points and extract richer feature points, a near-point dynamic collection system of adjacent points is set up to filter the noise points. The process is mainly divided into the following steps: (1) Select the node center point and take the point closest to the center point. (2) Establish the corresponding weights in each leaf node according to the number of points; then, the number of points from different leaves is dynamically extracted. (3) Set up the density measurement algorithm. When the number of target points extracted by each leaf remains unchanged, the density in a circle with a certain radius around each extraction point is measured. When the density is less than the threshold value, this point is removed and replaced by the next point until sufficient target points are extracted or all the points are looped. 

### 2.5. Cattle Body Size Measurement

The entire point cloud of the cow is segmented; then, key points based on different parts are located and calculated to eliminate interference from other parts. Figure 5 shows the measurement methods, in which the parameters are abbreviated as follows: body length (BL), withers height (WH), hip height (HH), thoracic circumference (TC), abdominal circumference (AC), and cannon circumference (CC).

#### 2.5.1. Body Height Measurement

To determine the cow’s WH and HH, (i) the front body point cloud and front leg point cloud and (ii) the back side point cloud and hind leg point cloud are used, respectively. The yellow areas are local extreme values selected from the upper part of the point cloud of the hind legs. Then, the midpoints $p_{0}$ and $p_{1}$ of the yellow areas and the midpoint $p_{c}$ of the two points are calculated. The plane parallel to the *Y*-axis is determined by $p_{c}$. A body slice A is obtained, and the point with the maximum *z* value in *A* is taken as the HH key point $p_{\mathrm{hip}}$. The vertical distance from $p_{\mathrm{hip}}$ to the ground is the cow’s HH.

#### 2.5.2. Circumference Measurement

The two front legs and the front body are used to determine TC. Then, $p_{2}$ and $p_{3}$ of the left and right front legs, respectively, are selected as the key points for measurement. The plane parallel to the *Y*-axis is determined by $p_{2}$ and $p_{3}$. A body slice B is obtained. The alpha shape is used to extract the measurements. The circumference comprises a point set $E=\{e_{1},e_{2},\ldots ,e_{n_{t}}\},(e_{i}=(xe_{i},ye_{i},ze_{i}),i=1,2,\ldots ,n_{t})$, and the $P_{\mathrm{thoracic}}$ of the tested cow is expressed as follows (Formula (3)); the following calculation is similar.
(3)$$P_{\mathrm{thoracic}}=\left\|e_{1}-e_{n_{t}}\right\|+\sum_{i=1}^{n_{t}-1}{\left\|e_{1}-e_{n_{t}}\right\|}_{2}$$

The mid-body area is used to determine AC. The smallest point $p_{4}$ of the Z-axis in the middle body is selected, and a plane is constructed from this point. The plane is extended along the *Y*-axis to obtain the body point cloud slice C, which forms the outline of AC. Considering the environment at the time of the shooting, one side of the body was shielded by the fence, resulting in incomplete point cloud data. Hence, a repair algorithm is proposed in this study. The proposed algorithm can reduce the calculation error caused by the absence of a unilateral point cloud. (1) The binary norm between all the contour key points $\left(\alpha_{x},\alpha_{y}),(\beta_{x},\beta_{y}\right)$ is calculated, and all the points between the farthest neighboring points are extracted to form set A. (2) The maximum point of the *Y*-axis is taken as the origin, and the axis of symmetry is parallel to the *Y*-axis. Moreover, set B is obtained by symmetrizing set A. (3) The distance to $\alpha_{x}$ is calculated by taking the minimum value of the *Y*-axis in set B as the endpoint. Then, the points in set B are shifted to α by the arithmetic value, resulting in a repaired point cloud.

#### 2.5.3. Cannon Circumference Measurement

CC is measured in the smallest dimension of the front leg of the cow’s body by extracting the boundary point in the contour of the leg point cloud. The two points with the closest *X*-axis distance are extracted, and the leg point cloud is sliced to form the outline of the CC. The linear interpolation algorithm is applied to find the point on the other side of the base side. 

#### 2.5.4. Body Length Measurement

The front and back of the body were used to determine BL. Moreover, the point cloud density algorithm proposed in this study to extract key points was used twice to extract the low-density regions and unwanted points at the edge of the point cloud. The threshold of the density was determined based on the location of the collected cattle point cloud data. After extraction, K-means clustering was conducted to cluster the shoulder-end cloud and the ischiatic-end cloud. Finally, two parts of point clouds D and E were obtained. The point $p_{5}$ with the smallest *x* value in D and the point $p_{6}$ in *E* were extracted as the key points, and the distance between the two points was the BL.

## 3. Results

### 3.1. Experimental Environment

In this study, a self-built dataset was used as the training and test data for the experiment. A total of 137 cattle of different breeds were collected in the pasture, and 150–200 frames of point cloud data were collected on each side of each cow with different body positions. The point cloud data were processed using translation, rotation, etc., for data augmentation. The dataset was constructed after selection. The data were divided into the training set, validation set, and test set in a ratio of 8:1:1. All training and testing were conducted using Windows 10 and A6000 devices. The programming software was Pycharm 2022.1.2, and the environment was Python 3.9.

### 3.2. Evaluation Indicators of Segmentation Results

Overall accuracy (OA) and mean intersection (mIoU) are used to evaluate performance. OA is the proportion of correctly segmented points relative to the total number of points (Formula (4)):(4)$$\mathrm{OA}=\frac{\sum_{i=0}^{k}p_{ii}}{\sum_{i=0}^{k}\sum_{j=0}^{k}p_{ij}}$$

The mean intersection (mIoU) is the ratio of the intersection size to the union size between the predicted and ground truth results, as follows (Formula (5)):(5)$$\mathrm{mIoU}=\frac{1}{k+1}\sum_{i=0}^{k}\frac{p_{ii}}{\sum_{j=0}^{k}p_{ij}+\sum_{j=0}^{k}p_{ij}-p_{ii}}$$

In Formulas (4) and (5), *k* + 1 is the number of classes. $p_{ii}$ represents the amount correctly predicted; $p_{ij}$ and $p_{ji}$ represent the number of false negatives and false positives, respectively. 

### 3.3. Cattle Body Point Cloud Segmentation Results

The proposed FPS [30], octree–PointNet++ [21], and DUOS were trained to compare the results. A comparison between the training results is shown in Figure 6. The proposed model achieved higher accuracy and mIoU in the training and testing sets than the other models. The model had good fitting ability and generalization. DUOS could provide more accurate segmentation results; it was better at performing the cattle body segmentation task, and it had the ability to meet the requirements of main point location in body size measurement. This is necessary for our application of cattle body segmentation, which can effectively improve the accuracy of subsequent body size calculations. The successful and unsuccessful segmentation results are displayed in Figure 7a–c. The segmentation results of the standard positions and the nonstandard positions of body bending and bowing are shown in (a) and (b), indicating that the model still maintains stability when applied to different positions. In a small number of the segmentation results, jagged edges or slight offsets appeared at the junction of the parts of the body, as shown in Figure 7c; this was due to registration errors and manual segmentation errors. Fortunately, these issues had little impact on the key point extraction. Given that this study aims to be applicable to cattle farms, it is essential to automatically segment the preprocessed cattle after field collection to achieve automatic body size measurement in animal husbandry. Figure 7d shows that the model can also complete the segmentation task well for unsegmented individuals, which has great advantages for practical applications. Furthermore, the model can accurately segment cattle data collected in the pasture. Figure 8 shows the comparison results between different models, and it is clear that DUOS has higher segmentation accuracy and smoothness.

### 3.4. Body Measurement Results 

The cattle in the standard position were chosen for manual body size calculation. We measured the real body size data of cattle in the same environment. Figure 9 shows that the same person measured the same body size parameters. To evaluate the quality of the measurement algorithm, manual measurement parameters were used as benchmarks. A comparison of our results with the two conventional algorithms is shown in Table 1, where RE represents the relative error between each parameter and the true value. Each parameter is compared to two traditional methods based on the overall point cloud, expressed as traditional first and traditional second. The purpose of the linear measurements is to locate the extreme and curvature key points under the whole point cloud. Circumference measurements are circumference calculations that locate extreme points to fit the surface and slicing algorithm under the whole point cloud [18,25,31]. The values in bold in the table indicate the results that are closer to the manual measurement results.

The average BL measurement error was 2.52%. The error in the result was affected by the shielding of the vertical fence. The error values of the height were lower because the point cloud of the back side was less occluded, and the calculation was easier. The segmented method could locate the key points more accurately, regardless of whether the parts were considered individually or jointly. The accuracy of the three body measurement parameters improved slightly, with average values of 2.12%, 2.26%, and 2.78%. The AC positioning was more accurate because the segmented part could find the key points of the abdomen more accurately, and the point cloud repair could effectively improve the calculation accuracy. The precision error of the CC was higher than that of the other parameters. As it is a new CC measurement method, there is still room for further research. The relative error of the measurement result depended on the needs of the ranch, but it did not exceed 5%. The results are acceptable. In general, the relative error of most of the cattle body size parameters measured using this method is closer to the true value than those measured using the whole point cloud and traditional key point measurement algorithms. 

## 4. Discussion

In the cattle segmentation experiment, compared with the traditional model, the proposed method is more accurate in terms of the mIoU parameters. The proposed model is robust in terms of adaptability to point cloud shape and noise processing. Thus, it was effectively applied to segmented cattle bodies. The model accurately divided the cow’s body into seven parts, which was very helpful for the subsequent experiments conducted to locate the key points of each part and measure the body size.

Regarding BL and CC, extracting the whole point cloud will yield many redundant points, affecting the subsequent clustering. The partitioned body part point cloud can completely eliminate the point cloud of the excess part, making it easier to locate the target area. When the maximum body is extracted directly, it will cause a significant deviation. Because of the different postures that affect the result, the distribution of the morphological features in the point cloud data will change. In contrast, after segmentation, the key points can be extracted according to the density value. The BL measurement can be accurately obtained by extracting the area of a certain density at the front of the shoulder and hip, which is less affected by the nonstandard stance of the cow. With the use of the proposed new measurement method, the combination of segmentation algorithms can make the calculation results more accurate. Hence, a method measuring the narrowest point is proposed; the method can be used to accurately locate the key point and measure the circumference, and the contour point extraction algorithm can effectively reduce the calculation time without affecting the calculation accuracy. The segmented data of the front leg can be directly selected without considering the impact of the hind leg. Regarding AC, when the cattle are in a nonstandard position, the body shows a C shape at the top of the viewing angle, affecting the positioning accuracy. Instead, our method requires only a slice of the abdominal area and does not need to consider the other parts. Regarding body height, the midpoint of the two front legs can be used to avoid the key point deviation problem caused by the legs being in a nonstandard position. Thus, this method is different from the methods based on the whole point cloud, which are used to directly find the extreme curvature value of the back side.

## 5. Conclusions

To solve the problem of the inaccurate positioning of key points in the automatic measurement of cattle body sizes based on the whole point cloud data, a measurement method based on the segmentation results of the cattle body point cloud is proposed. Compared with other traditional algorithms, the proposed DUOS algorithm improved the sampling and grouping methods. The proposed method can provide better results in the cattle point cloud data: 92.87% in OA and 85.91% in mIoU. Compared to octree–PointNet++ and PointNet++, the model improved the results by 0.53% and 1.21%, respectively. Moreover, the proposed method can automate the body size measurement of parts using body size segmentation. The algorithm has higher accuracy and practicability than other methods based on the whole point cloud. Specifically, secondary local density detection and clustering algorithms are used for BL. For TC and AC, the key points are collected according to the segmented point cloud, from which the point cloud slice is extracted. The key point compensation algorithm is used to deal with unilateral point cloud voids caused by the AC barrier. For CC, a new method for measuring the minimum distance of the corresponding points is proposed to find the thinnest part. The experimental results of the cattle body size measurements show that, compared with the whole point cloud measurement method, the proposed method based on point cloud segmentation is more consistent with the actual measurement results. The study had some limitations. Computational correction methods were used to reduce the measurement errors, but the errors still exist. The data sample size is small. We will try to address these issues in future studies. Therefore, the proposed DUOS–PointNet++ model can better complete the cattle body size segmentation task. The research can be applied to body size measurement, and it can also be extended to weight estimation or to scoring the body condition of individual parts. Additionally, it can be applied to other medium- and large-sized livestock; thus, it has good prospects and applicability in the development of intelligent animal husbandry.

## Figures and Tables

**Figure 1 animals-14-02553-f001:**
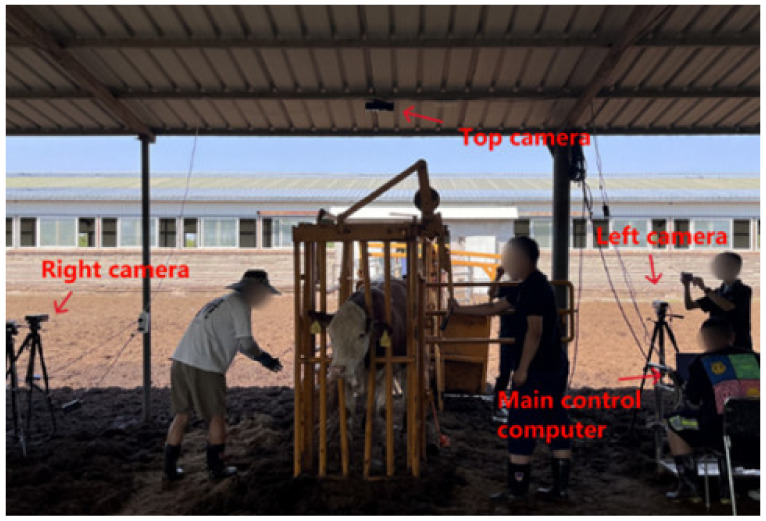
Experimental data acquisition site.

**Figure 2 animals-14-02553-f002:**
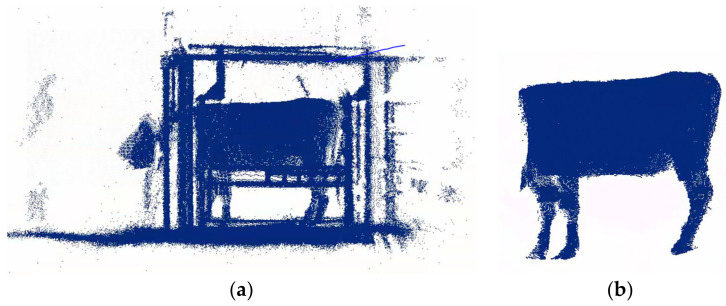
Point cloud preprocessing: (**a**) registered point cloud; (**b**) preprocessed point cloud.

**Figure 3 animals-14-02553-f003:**
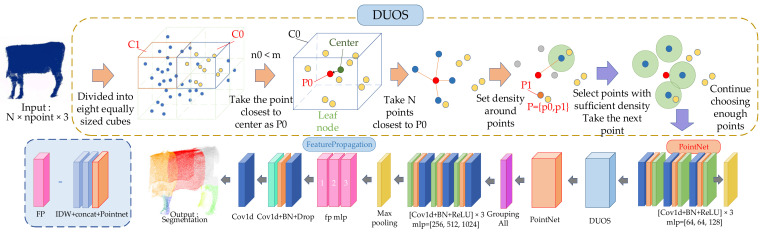
Specific architecture of improved PointNet++ segmentation network based on DUOS.

**Figure 4 animals-14-02553-f004:**
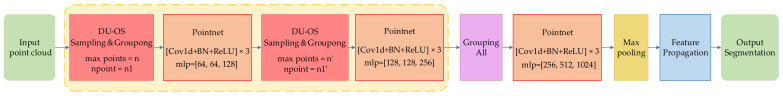
Algorithm flow chart.

**Figure 5 animals-14-02553-f005:**
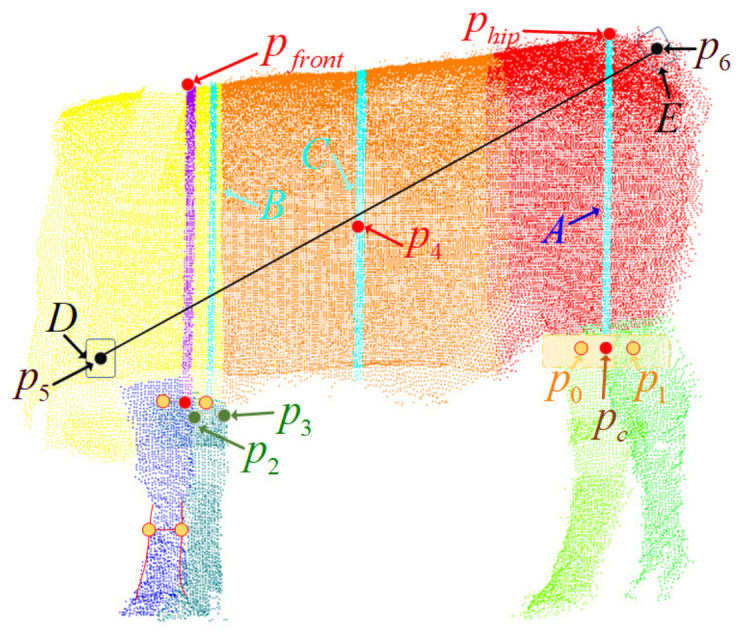
Schematic diagram of body size measurement.

**Figure 6 animals-14-02553-f006:**
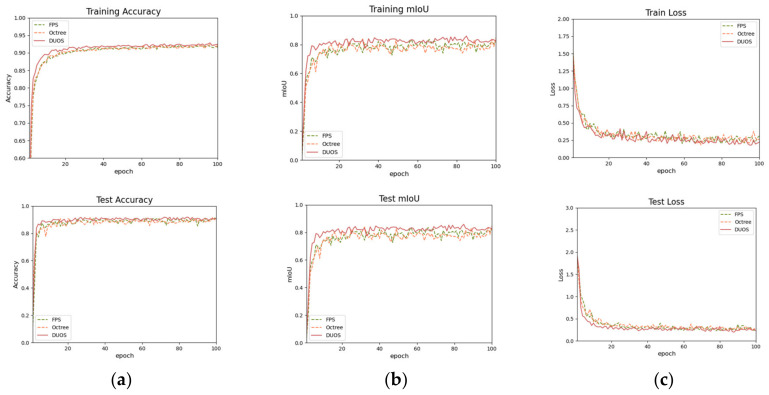
Comparison of the results of training sets and test sets: (**a**) Acc; (**b**) mIoU; (**c**) loss.

**Figure 7 animals-14-02553-f007:**
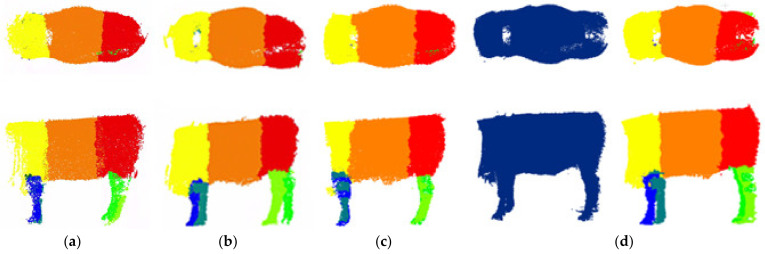
Segmentation results of networks: (**a**) standard posture; (**b**) nonstandard posture; (**c**) unsuccessful segmentation; (**d**) undivided individual.

**Figure 8 animals-14-02553-f008:**
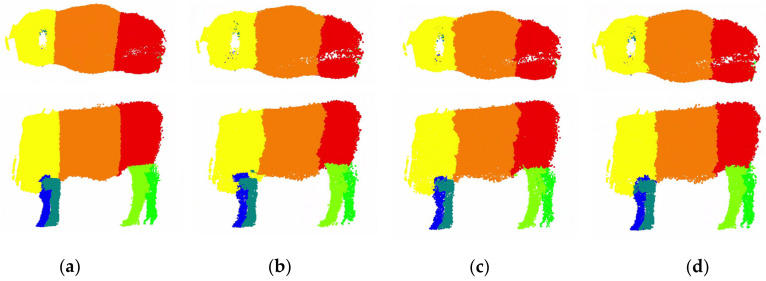
Segmentation results comparison: (**a**) manual segmentation; (**b**) FPS segmentation; (**c**) octree segmentation; (**d**) ours.

**Figure 9 animals-14-02553-f009:**
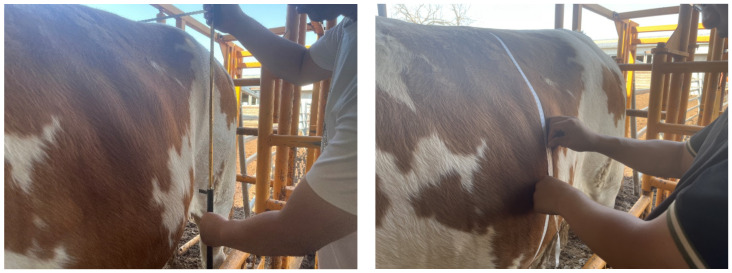
Manual measurement of cattle body size parameters.

**Table 1 animals-14-02553-t001:** Comparison between the proposed method and traditional measurement methods.

**Cattle**	**Withers Height**					**Hip Height**				
	**Manual**	**WF**	**WS**	**Ours**	**RE**	**Manual**	**WF**	**WS**	**Ours**	**RE**
1	125	120.65	125.98	**124.22**	**0.62%**	132.5	130.81	134.32	**133.33**	**0.63%**
2	143	140.50	141.22	**142.32**	**0.48%**	145.5	144.23	147.10	**144.45**	**0.72%**
3	138.5	136.86	140.76	**138.01**	**0.35%**	146.5	145.10	150.57	**147.74**	**0.85%**
4	134	133.21	132.34	**133.35**	**0.49%**	136	130.27	138.56	**137.21**	**0.89%**
5	131.5	132.44	135.27	**131.91**	**0.31%**	134.2	136.88	139.77	**135.99**	**1.33%**
6	146	148.18	148.37	**146.34**	**0.23%**	150	147.80	154.79	**151.87**	**1.25%**
7	131.5	134.58	132.93	**130.12**	**1.05%**	134.2	140.02	141.34	**135.21**	**0.75%**
8	125.5	130.62	130.65	**127.16**	**1.32%**	132	134.46	139.21	**133.28**	**0.97%**
9	138	135.24	139.16	**138.79**	**0.57%**	142	146.94	151.70	**144.15**	**1.51%**
10	140	138.92	141.02	**139.21**	**0.56%**	146	150.61	153.76	**148.28**	**1.56%**
**Cattle**	**Body Length**					**Thoracic Circumference**				
	**Manual**	**WF**	**WS**	**Ours**	**RE**	**Manual**	**WF**	**WS**	**Ours**	**RE**
1	139.5	142.46	134.12	**139.70**	**0.14%**	188	177.68	187.56	**187.63**	**0.20%**
2	165	168.51	168.49	**163.05**	**1.18%**	216	231.86	224.48	**220.57**	**2.12%**
3	173.5	176.26	169.91	**171.11**	**1.38%**	217	207.22	**216.95**	**216.95**	**0.02%**
4	173.5	170.15	170.66	**171.20**	**1.33%**	178	178.95	176.62	**178.80**	**0.45%**
5	150	153.12	146.11	**151.47**	**0.98%**	206	191.23	200.25	**205.25**	**0.36%**
6	170	166.89	164.32	**168.62**	**0.81%**	220	242.56	218.54	**222.28**	**1.04%**
7	149.5	140.09	147.74	**148.48**	**0.68%**	203	190.48	191.08	**206.79**	**1.87%**
8	146	138.55	139.24	**142.61**	**2.32%**	181	169.15	176.21	**176.36**	**2.56%**
9	173	168.20	166.96	**169.79**	**1.86%**	230	225.44	233.49	**234.40**	**1.91%**
10	151	159.94	152.94	**152.10**	**0.73%**	205	**204.85**	195.07	198.11	3.36%
**Cattle**	**Abdominal Circumference**					**Cannon Circumference**				
	**Manual**	**WF**	**WS**	**Ours**	**RE**	**Manual**	**WF**	**WS**	**Ours**	**RE**
1	219	215.16	200.38	**215.93**	**1.40%**	19.5	22.31	**18.36**	**18.36**	**5.85%**
2	257	266.92	**255.66**	**255.66**	**0.52%**	21	24.05	**20.56**	**20.56**	**2.10%**
3	258	263.09	253.35	**257.40**	**0.23%**	22	23.29	24.41	**22.79**	**3.59%**
4	211	242.98	**211.45**	**211.45**	**0.21%**	18	20.57	**18.97**	**18.97**	**5.39%**
5	241	246.54	224.94	**238.53**	**1.02%**	20	19.93	22.85	**20.06**	**0.30%**
6	256	265.46	247.32	**252.20**	**1.48%**	23	23.46	21.09	**23.23**	**1.00%**
7	233	221.00	224.36	**230.67**	**1.00%**	20	23.75	19.03	**19.60**	**2.00%**
8	234	205.27	221.62	**234.88**	**0.38%**	20	20.88	21.44	**19.42**	**2.90%**
9	257	254.98	252.31	**258.48**	**0.58%**	24	26.38	26.07	**25.32**	**5.50%**
10	248	235.74	238.50	**240.06**	**3.20%**	20.5	22.00	**19.76**	**19.76**	**3.61%**

The values in bold indicate the results that are closer to the manual measurement results.

## Data Availability

The authors do not have permission to share the data.

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
