# Peer review of "Cattle Body Size Measurement Based on DUOS–PointNet++"

_animals, 2024, doi:10.3390/ani14172553_

Round 1

Reviewer 1 Report

Comments and Suggestions for Authors

Review comments on “Cattle body size measurement based on DUOS-PointNet++” by Zhi Weng etl.

This work presents a new algorithm named dynamic unbalanced octree grouping (DUOS) based on PointNet++ and an efficient method of body size measurement based on segmentation results.

My main general comments are as below:

- The article needs more elaboration about: outcomes, limitations, and possible/future scenarios.

- An important shortcoming is that the author does not highlight the contribution of their manuscript in comparison to the work that has been performed by previous researchers. This can be added in the introduction and/or conclusion section.

- The work will be significant if the database, the source codes, and the proposed model are presented to the public for a detailed analysis of the proposed model.

- The authors should investigate the stability of the proposed model because image can be degraded by additive noise, in the presence of cluttering backgrounds, geometric modifications such as pose changing and scaling, nonuniform illumination, and eventual object occlusions.

- Why didn't the authors use image normalization and pre-processing to improve the proposed model? The subsection 2.2 should be expand.

- The authors didn’t provide a comparison of the performances on training and testing sets. The authors should investigate experimentally the overfitting of the proposed model.

- The authors should provide the example of unsuccessfully segmentation. The authors should investigate the reason.

- The authors should provide critical analysis of the results of statistical analysis. It might have been enough to provide confidence intervals or percent deviation.

- The authors use a small sample size of the animals. The authors do not use data augmentation.

- The authors should make the method steps from 2.4.1 into a flow chart.

-An important question remains. When capturing, the animal is in a cage, which occludes the animal from the camera. This leads to an incomplete cloud. Are there any methods used to inpainting the incomplete cloud?

- It is necessary to compare the proposed model with the latest research methods to highlight its advantages. For example:

Ruchay, A., Kober, V., Dorofeev, K., Kolpakov, V., Miroshnikov S.  Accurate body measurement of live cattle using three depth cameras and non-rigid 3D shape recovery (2020) Computers and Electronics in Agriculture, 179, 105821.

Du A., Guo H, J. Lu J., Su Y., Ma Q., Ruchay A., Marinello F., Pezzuolo A. Automatic livestock body measurement based on keypoint detection with multiple depth cameras (2022) Computers and Electronics in Agriculture, 198, 107059.

Reviewer 2 Report

Comments and Suggestions for Authors

1.In the introduction section, the statement "At present, most of the cattle body size measurements based on point cloud use the complete cattle point cloud after registration" could be supported with specific references to related studies. Please enumerate these studies and analyze their specific advantages and disadvantages. Additionally, apart from point clouds, consider discussing whether depth images have been used for similar work.

2.The imaging quality of depth cameras is significantly affected by lighting conditions, especially since the data collection in this study was conducted outdoors. It would be beneficial to present some raw point clouds and the processing steps applied to them to demonstrate the data quality. This is crucial as the accuracy of key point localization largely depends on the quality and completeness of the point cloud data from the relevant body parts.

3.The paper lacks comparative experiments with other studies in terms of body size measurement and point cloud segmentation. Including such comparisons would provide a clearer picture of the proposed method's performance relative to existing approaches.

Comments on the Quality of English Language

The author uses a lot of long sentences, which seems to be laborious, and I think that where you can cite, you can directly cite the literature without much explanation.

Reviewer 3 Report

Comments and Suggestions for Authors

This paper is very interesting, and looks suitable for publication. This reviewer suggest the next minor corrections:

1. Authors should explain why they are using Kinect V2 instead of any other camera or vision's system.

2. Authors should explain why they are using RANSAC instead of any other method, like PROSAC.

3. ¿How the accuracy can be improved? ¿By another camera? ¿By implementing another method?

4. In Table 1, the authors should add the obtained error in each case.

5. ¿What kind of error are the authors obtaining? ¿Absolute error?

6. ¿Is the obtained error acceptable? 

Round 2

Reviewer 1 Report

Comments and Suggestions for Authors

All suggestions and comments were correted